# Factors Associated with Postpartum Maternal Functioning in Black Women: A Secondary Analysis

**DOI:** 10.3390/jcm12020647

**Published:** 2023-01-13

**Authors:** Kortney Floyd James, Betsy E. Smith, Millicent N. Robinson, Courtney S. Thomas Tobin, Kelby F. Bulles, Jennifer L. Barkin

**Affiliations:** 1National Clinician Scholars Program, David Geffen School of Medicine, University of California, Los Angeles, CA 90024, USA; 2Department of Internal Medicine, Mercer University School of Medicine, Macon, GA 31207, USA; 3School of Social Work, University of North Carolina at Chapel Hill, Chapel Hill, NC 27516, USA; 4Los Angeles Jonathan and Karin Fielding School of Public Health, Department of Community Health Sciences, University of California, Los Angeles, CA 90095, USA; 5School of Medicine, Department of Community Medicine, Mercer University, Macon, GA 31207, USA

**Keywords:** maternal functioning, postpartum mental health, Black motherhood

## Abstract

In the United States, 29–44% of Black women experience postpartum depressive symptoms (PDS), yet few are properly identified and/or connected to mental care services. The purpose of this secondary analysis was to examine the relationship between maternal functioning and clinical variables (PDS, maternal–infant attachment), racial variable (Black racial identity types—low race salience, assimilated and miseducated, self-hating, anti-White, multiculturalist, and conflicted), and sociodemographic characteristics (relationship status, education, insurance, childbirth type). A total of 116 women living in the southern United States were included in the analysis. Multivariate analyses revealed that Black racial identity (*p* = 0.02), PDS (*p* < 0.0001), maternal–infant attachment (*p* < 0.0001), and educational level (*p* = 0.03) were independently associated with maternal functioning. This work provides new evidence regarding the role of various clinical and racial factors on Black postpartum women’s adjustment to motherhood. This analysis also adds to the growing body of evidence of reliability for the BIMF in Black postpartum women.

## 1. Introduction

In the United States (US), 29–44% of Black women experience postpartum depressive symptoms (PDS) [1,2]. Despite this high percentage of Black women experiencing symptoms, few are properly identified as having postpartum depression and receive mental health services [2,3]. The reasons for this disparity are unclear but likely multi-factorial. The intersection of Black women’s gender, race, racial identity, and class make them susceptible to various types of discrimination, including sexism, racism, and classism, which increase their mental and physical stress [4,5], possibly increasing their risk of experiencing PDS. Severity of PDS may also impact maternal functioning, a woman’s ability to care for herself and infant, and resume their typical activities prior to giving birth [6]. Previous work has examined the relationship between Black racial identity and PDS [1] or maternal functioning [7] individually but has not considered how they collectively influence maternal functioning. In this analysis, we employ the triangulation of concepts from the Nigrescence Theory [8] and the Becoming a Mother (BAM) theory [9] as a framework to guide our selection and analysis of PDS, social (e.g., education, income, relationship status) and racial factors (Black racial identity) that may influence Black women’s functional status after childbirth.

### 1.1. Theoretical Framework: The Nigrescence Theory and Becoming a Mother Theory

Mercer’s BAM theory describes the adjustments a woman begins to make in her life after learning that she is pregnant through the postpartum period [9]. While the BAM theory considers various factors that may influence the journey into motherhood [9], influences unique to Black women are not considered. Much of Mercer’s research focused on pregnant and postpartum White women who were affluent, highly educated, and partnered or Black women who were young, single, teenaged mothers of lower socioeconomic status [10]. This narrow perspective of Black women’s journey in motherhood does not consider the influence that being Black in a racialized society may have on one’s mental health during that time. For this analysis, the BAM theory was adapted to include Black racial identity, a construct from the Nigrescence Theory. The Nigrescence theory describes the racial identity of Black people who live in the US–a racialized society. Racial identity is defined as the significance and meaning of self that individuals attribute to their membership within Black racial groups [8]. Black racial identity has been associated with psychological distress (e.g., depressive symptoms) in Black adults, but not during the postpartum period. 

The concepts of maternal functioning and maternal mental well-being (the presence or absence of PDS) are essential in describing the journey of becoming a mother and was examined in the study. While the concepts (i.e., maternal functioning, PDS, Black racial identity) are centered around the mother, they collectively influence the maternal–infant bond [9]. The bond developed between the postpartum woman and infant is established when the mother recognizes and responds to her infant’s cues. By appropriately responding to the infant’s needs, the infant’s attachment/security is also nurtured [9]. This theoretical adaptation guided this analysis to allow the proper selection of constructs and the relationships between them. 

### 1.2. Maternal Functioning

Maternal functioning is a woman’s ability to care for herself and infant and reflects the quality of life a woman has as a new mother [6,11]. Assessing maternal functioning and PDS may help identify Black women experiencing difficulties adapting to motherhood [6]. The Barkin Index of Maternal Functioning (BIMF) was developed in 2010 [12], [13] to further understand and quantify maternal functioning of women up to one year after childbirth. The BIMF focuses on a woman’s judgment of her self-care, infant care, bond with her infant and adjustment to motherhood, while handling other various responsibilities, and adequacy of their support system [14]. The BIMF has been translated, adapted, and validated for use in Iran, Turkey, and Ethiopia [15,16,17,18]. The BIMF has also been used in Australia to assess the effectiveness of support groups in decreasing women’s likelihood to experience perinatal depression and difficulties adjusting to motherhood [19]. 

While this international utilization supports the BIMF’s reliability across populations, this analysis will focus on variables that are predictive of or associated with the functioning abilities of postpartum Black women living in the United States. In qualitative interviews, Black women reported that social support, their child’s temperament, self-care, maintaining a life and identity outside of being a mother, and firsthand experience parenting influenced their adjustment to motherhood [20]. Maternal functioning abilities have been inversely (and significantly) associated with the emotional symptoms of anxiety [21], depression [22,23] in Black postpartum women. Further, previous studies presented correlations between maternal functioning abilities and sociodemographic characteristics that seem counterintuitive. For instance, unpartnered Black women with lower levels of education reported higher maternal functioning abilities in comparison to partnered Black women with higher levels of education [21]. In several studies, single women reported higher levels of functioning in comparison to partnered or married women [21,22]. Additionally, maternal functioning was found to decrease as women’s age increased [13], yet was higher in those with government-funded health insurance in comparison to those with employer-sponsored health insurance [6]; the samples from these studies were described as White or non-White, so it is unclear if these findings were evident in Black women. Although the BIMF provides a person-centered approach to assess psychological well-being and functioning abilities of all postpartum women, the focus of this study necessitated the inclusion of other factors that may correlate with Black women’s PDS and adjustment to motherhood.

### 1.3. Black Racial Identity

To consider a characteristic that is unique to Black women’s maternal functional status, Black racial identity was examined. Black racial identity refers to the importance and value that Black people living in America attribute to their racial group membership [24]. Prior literature suggests that Black people who have a positive regard towards being Black (e.g., Afrocentric or Multicultural racial identity types) have healthier psychological well-being when compared to those who do not (e.g., Miseducated and Assimilated or Self-Hating racial identity types) [1,25]. While there is growing evidence supporting the influence of Black racial identity on pregnancy-related outcomes [1,6], there is a lack of research examining its relationship to the bond shared between a mother and their infant.

### 1.4. Postpartum Depressive Symptoms

Postpartum depression, which is the most common complication of childbirth [2], is a mood disorder that may develop after childbirth and persist or worsen if not treated. PDS can range in severity; some women may doubt their abilities to care for their infant or withdraw from family and friends, while other women may have thoughts of harming themselves or their infant [26]. The severity and duration of PDS can influence a woman’s overall health, maternal functioning, and the mental health and development of her infant [2]. Due to the symbiotic relationship between a woman and her infant, mental well-being and functioning abilities may influence a woman’s attachment with her infant as well. 

### 1.5. Maternal–Infant Attachment

Maternal–infant attachment refers to the deep, emotional bond or connection between a mother and her infant through the first year of the child’s life [27]. A woman’s attachment to her infant begins immediately after childbirth and is enhanced when the infant is placed skin-to-skin with the mother; the bond is then further developed through feeding and other activities performed to care for the infant [28]. Attachment is not only an emotional connection, but literature has shown that it has long-term effects on the infant’s socialization, security, and well-being [29]. Additionally, attachment has a protective effect against the development of postpartum depression [30]. Positive, secure maternal–infant attachment develops as the mother addresses the infant’s needs on a consistent basis, which develops the trust the infant has in others and eventually, their positive feelings of self-worth [29]. 

### 1.6. Study Purpose and Aim

The relationship between Black racial identity, postpartum depression [1] and maternal functioning [7] have been examined separately. Both analyses involved assessing Black racial identity alone and did not account for any other factors or the association between postpartum depression and maternal functioning. The primary aim of this analysis was to expand the work previously conducted with the Black racial identity clusters by examining the relationship between maternal functioning and the Black racial identity clusters with the addition of clinical variables (PDS, maternal–infant attachment), and sociodemographic characteristics (relationship status, education, insurance, childbirth type). 

## 2. Materials and Methods

This secondary data analysis was performed using data from a cross-sectional study designed to identify racial identity clusters and examine the clusters’ relationships to the mental well-being of Black postpartum women living in Georgia. The original study was conducted in accordance with the Declaration of Helsinki and approved by the Institutional Review Board of Georgia State University (protocol code H20409 and date of approval 6 February 2020). Additionally, informed consent was obtained from all subjects involved in the original study. Details regarding the original study have been published elsewhere [1]. As a secondary analysis of de-identified data, this study was determined to be exempt research by the IRB at the University of California, Los Angeles (IRB# 22-001912).

### 2.1. Sample and Participants

To be considered eligible, women had to (a) self-identify as Black/African American, (b) be 18 years or older, (c) have an infant 4 weeks to 12 months of age, (d) and speak English. Participants could also not have factors that increased their likelihood to experience postpartum depression, which included a history of previously diagnosed mental illness including depression, extended separation from their infant due to hospitalization for medical problems, birth of multiples, or other children having chronic health conditions [1]. Potentially eligible women were recruited from February through April 2020 in Atlanta, Georgia, and surrounding counties, and resulted in a convenience sample size of 116.

### 2.2. Measures

#### 2.2.1. Outcome Variable

The BIMF was used to measure maternal functioning. The instrument was developed in 2010 to assess functional status and psychological well-being of postpartum women [12]. The BIMF has 20 items on a Likert scale with responses ranging from “0”–strongly disagree to “7”–strongly agree; two items, 16 and 18 are reverse coded (“I worry about how other people judge me as a mother,” and “Anxiety or worry often interferes with my mothering ability”). The BIMF also includes items such as, “I feel rested”, “I am taking good care of my baby’s physical needs (feedings, changing diapers, doctor’s appointments)”, and “I am a good mother.” Total scores range from 0 to 120, with higher scores indicating higher maternal functioning. Through focus groups, the instrument developers determined that seven functional areas must be addressed to quantify maternal functioning: (1) self-care; (2) infant care; (3) mother-child interaction; (4) psychological well-being; (5) social support; (6) management; and (7) adjustment [12]. Although some items in BIMF measures psychological well-being, these items are different than those in commonly used tools that assess postpartum mental status because BIMF emphasizes the woman’s ability to complete tasks related to caring for herself and her infant. While the BIMF includes the domain of psychological well-being, it is within the context of mothering. The Cronbach’s alpha for the BIMF in this sample of Black women is 0.84, reflecting high reliability. 

#### 2.2.2. Exposure Variables

The CRIS was used to identify the Black racial identity clusters [31] within the sample. The CRIS is a 40-item questionnaire with seven-point Likert scale (“1” strongly disagree to “7” strongly agree) that captures the complexity of Black racial identity, which includes items such as, “As an African American, life in America is good”, “I hate the White community and all that it represents”, and “I sometimes struggle with negative feelings about being Black.” The CRIS produces scores for six subscales: assimilation, miseducation, self-hatred, anti-white, Afrocentricity, and multiculturalist. Scores for each subscale range from 1 to 7 with higher scores reflecting greater endorsement of the attitudes measured in each subscale. Black racial identity clusters for the sample were previously identified by performing hierarchical cluster analysis of participants’ CRIS subscale score [1]. They include Low Race Salience, Assimilated and Miseducated, Self-Hating, Anti-White, Multiculturalist, and Conflicted. In previous research, the Cronbach’s alpha of the CRIS subscales ranged from 0.70 [32] to 0.91 [33] in samples of Black individuals, including women. In the original study, the CRIS sub scales had high reliability (α = 0.79 − 0.87) [1].

The EPDS was used to measure postpartum depressive symptoms. The EPDS is a 10-item questionnaire that measures psychological symptoms of postpartum depression [34] through questions like “I have been able to laugh and see the funny side of things” and “the thought of harming myself has occurred to me”. Each item has four possible responses which range from: “no, not at all” to “yes, very often:”; or “no, not at all” to “yes, quite a lot”; or “as much as I ever did” to “no, not at all”; or “yes, most of the time I haven’t been able to cope at all” to “no, I have been coping as well as ever”. Total scores range from 0 to 30 with higher scores indicating greater depressive symptoms. A score of 10 or higher may indicate depression and warrants further clinical assessment [34]. Using a cut-off score of 10, the sensitivity and specificity of the EPDS ranges from 85% to 84%, respectively [35], both of which are adequate. In this study, the EPDS scale had high reliability as reflected by a Cronbach’s alpha of 0.89.

Maternal–infant attachment was measured using the MAI [36]. The MAI consists of 26 items with a 4-point Likert scale with questions such as, “I feel love for my baby” and “I understand my baby’s signals.” Response options range from “almost always” to “almost never”; total scores range from 26 to 104 with higher scores reflecting higher levels of affectionate maternal–infant attachment. The MAI has been used in studies which included postpartum women and their infants of various ethnicities and nationalities and has been translated to multiple languages, including Korean and Portuguese [37]. However, the MAI has not been validated in samples with Black mothers. In previous studies the MAI had adequate to high reliability (Cronbach’s alpha 0.76 to 0.93) in samples of White/European mothers and their infants throughout the first year of the postpartum period [38]. The tool also had an adequate Cronbach’s alpha of 0.76 to 0.85 in samples with Taiwanese mothers whose infants were 4 weeks to 8 months of age [28]. In this study the Cronbach’s alpha was 0.93.

A survey that consisted of 12 multiple choice items was used to describe the demographic characteristics of the sample. This self-report survey captured participants’ age, the age of their youngest infant, total number of children, method of childbirth for youngest infant (vaginal or cesarean), employment status irrespective of family and medical leave (unemployed, part-time, full-time), healthcare insurance (government funded, private, none), highest level of education (did not complete high school, high school diploma or General Education Diploma (GED), associate or technical degree, college or graduate degree), relationship status (single or married/living with partner), and income level (less than USD 26,000, USD 26,000–USD 74,999, or USD 75,000 and greater).

### 2.3. Statistical Analyses

Characteristics of the sample were described with numbers and percentages for categorical variables and means (with standard deviations) and medians (with ranges) for continuous variables. Associations between maternal functioning and the racial identity clusters, as well as additional characteristics, were assessed with both bivariate and multivariate analyses. Bivariate analyses included the Mann–Whitney U or Kruskal–Wallis test for assessing categorical measures. The Dwass, Steel, Critchlow–Fligner multiple comparison test was used for post hoc comparisons with significant results from the Kruskal–Wallis test. Spearman’s rank-order correlation was used for assessing continuous measures. Multivariate analyses were conducted with linear regression. Dummy variables were created for variables that had more than two categories. All variables included in the bivariate analyses were included in the multivariate analyses regardless of their significance status in the bivariate analyses. This was to account for any variable that may be non-significant in the bivariate analysis but could become significant when adjusting for other variables. In other words, we were controlling for potential confounding variables with this analytic approach. Variables were entered into the linear regression model using the forward selection method. This involves adding one variable at a time to the model, and only the variables that improve the fit of the model and meet the specified level of significance are retained [39]. Statistical significance was determined with a *p* < 0.05, and all analyses were performed with SAS^®^ Statistical Software, Version 9.4 [39] (SAS Institute, Cary, NC, USA).

## 3. Results

### 3.1. Sample Characteristics

Previous reporting of the sociodemographic characteristics of this sample of Black postpartum women (*n* = 116) revealed that the ageof the women ranged from 18 to 41 years (M = 29.5; SD = 5.3) and their infants were 1 to 12 months old (M = 5.6; SD = 3.5) [1]. The majority of women were married or cohabitating with their partner (71%), had a college degree or higher (47%), worked full-time (57%), had given birth to their youngest infant vaginally (62%), had government funded insurance (55%), and a total household income of USD 26,000 or more (65%) [1]. Additionally, it was previously reported that the most populous Black racial identity cluster was Multiculturalist (*n* = 42, 36%) cluster, and the least populous cluster was Low Race Salience (*n* = 5, 4.3%) [1]. Overall, the women in the sample had a high level of maternal functioning (median = 98.0), low levels of PDS (median = 7.0), and an affectionate attachment to their infant (median = 103.0) (Table 1).

### 3.2. Bivariate Analyses and Construct Validity of the BIMF

Bivariate associations are reported in Table 2. Although previously published, the associations and post hoc analyses between the racial identity clusters and maternal functioning (*p* = 0.0012) [7] to provide context with what variables were considered for multivariate analyses in regard to the main exposure (i.e., racial identity clusters). Maternal functioning was significantly, negatively correlated with the EPDS score (rho = −0.48, *p* < 0.0001) but significantly, positively correlated with the MAI score (rho = 0.38, *p* < 0.0001) [40,41]; these significant correlations also provide further evidence of construct validity for the BIMF. Specifically, one would expect maternal functioning to increase in relation to a decrease in depressive symptomology and this occurred as expected. Likewise, the positive, significant correlation indicates that the BIMF and MAI scores are interacting in a manner consistent with expectation. The BIMF scores were significantly different by level of education (*p* = 0.01). Post hoc analyses showed that the BIMF score of mothers who completed a high school/GED education or less (median = 106.0) was different from the BIMF score of mothers who had completed a Bachelor’s degree or higher (median = 96.0) and the BIMF score of mothers who had completed some college/tech school/associate’s degree (median = 95.0).

### 3.3. Multivariate Analyses

Multivariate analyses revealed that racial identity clusters (*p* = 0.0238), EPDS score (*p* < 0.0001), MAI score (*p* < 0.0001), and level of education (*p* = 0.0342) were significantly associated with maternal functioning (Table 3). On average, the BIMF score for mothers in the Self-Hating cluster was 11 points lower than the BIMF score for mothers in the Assimilated and Miseducated cluster. The BIMF score for mothers in the Anti-White cluster was almost 8 points lower than the BIMF score for mothers in the Assimilated and Miseducated cluster. On average, mothers who completed some college/tech school/associate’s degree had a BIMF score 6.7 points lower than mothers who completed a high school/GED education or less. The EPDS score was associated with diminished maternal functioning, while the MAI score was associated with increased maternal functioning.

## 4. Discussion

The purpose of this secondary analysis was to identify clinical, racial, and sociodemographic factors associated with maternal functioning both independently and in conjunction with each other in a population of Black postpartum women. To our knowledge, this analysis is the first to examine the association between racial identity, PDS, maternal–infant attachment, and education level in postpartum Black women using an adjusted model. Additionally, this analysis adds to the growing body of evidence of reliability for the BIMF and is the first to demonstrate reliability in an exclusively Black population of women living in the US. Racial identity clusters, PDS as measured by the EPDS, maternal–infant attachment, and level of education were significantly associated with maternal functioning. Multivariate models revealed these factors explained more than 50% of the variation in maternal functioning levels within the sample, reflecting that these factors are salient to the functioning of postpartum Black women. Because the relationship between Black racial identity and maternal functioning have been published elsewhere [7], we will discuss the remaining findings. 

### 4.1. Maternal Functioning and Postpartum Depressive Symptoms

The postpartum women in this sample had an overall low level of PDS and thus, a high level of maternal functioning. Previous research found that young adult Black women who are experiencing distress tend to face a lower risk for depression [40]. The present study had similar characteristics in regard to age (mean age = 29.5) and low level of PDS (mean EPDS score = 7). Culturally, Black women may rely on the “Strong Black Woman” typology or Superwoman Schema to balance the stress of motherhood without becoming overwhelmed [41]. As a result, these women may experience distress without compromising their abilities to respond to the needs of their infant/child. While the Superwoman Schema may empower some women and foster resilience despite hardships [42], others may suppress their emotions, avoid expressing difficulty adjusting to motherhood, and influence their help-seeking behaviors [41,43]. The barriers to seeking mental health services may be further exacerbated by the biases that perinatal care providers have (e.g., making inappropriate assumptions, invalidating concerns), which impacts the care Black women receive and the mental health services they may be offered [3,41]. Studies have shown that the maternal experiences of Black women are challenging due to the accumulative stress of living in a racialized society with intersecting systems of oppression (e.g., gendered racism, ageism, sexual identity) that negatively influence their emotional well-being, mental health [2,5,40] and consequently–their maternal functioning abilities. Therefore, the link between PDS and maternal functioning are expected because these two components are empirically related to one another [44]. 

### 4.2. Maternal Functioning and Maternal–Infant Attachment

On average, the participants in this study had high MAI scores, which indicated a high level of affectionate attachment. This finding was consistent with the attachment scores of Turkish mothers (M = 100.1) in an interventional study who received eight weeks of prenatal education about various topics related to parenthood (e.g., attachment to their infant, physical/emotional postpartum changes) [45] but higher than the mean MAI scores of Korean postpartum women whose babies had been admitted into a neonatal intensive care unit (M = 85.6) [46]. This suggests that the women in this study had not been separated from their infants for an extended period, that they were knowledgeable about caring for an infant, and attentive to their infant’s needs. Lastly, due to the long-term implications that attachment type has on the social interaction of the infant [30], it is possible that maternal functioning levels may be useful in identifying mother-infant dyads in need of resources to support the woman’s adjustment to parenthood and enhance their bond.

### 4.3. Maternal Functioning and Level of Education

In this study, Black women who had a high school education/GED or less had higher levels of maternal functioning in comparison to those with post-secondary education, a finding that has been found in a previous study examining maternal functioning levels of minoritized and marginalized women [21]. The Black cultural phenomenon Sojourner Syndrome may provide insight into this seemingly counterintuitive relationship. Sojourner Syndrome, as named by Dr. Mullings after the once enslaved abolitionist and women’s rights activist Sojourner Truth, describes Black women’s ability to persevere despite the negative impact of intersecting oppressions (e.g., classism, racism, sexism) on their health and well-being [4]. Mullings’ research in Harlem found that Black people with lower socioeconomic status who lived in harsh, under-resourced, predominantly Black neighborhoods found positive, protective aspects within their neighborhood [4]. The Harlem residents were happy in their neighborhoods due to the collectivism within the Black community, spiritual support, cultural belonging, and minimal exposure to racism, all of which served as protective factors for their emotional well-being. It is possible that the women in this study with lower levels of education may also live in similar impoverished neighborhoods, yet also find comfort and support within their community, resulting in higher levels of maternal functioning when compared to those with higher levels of education. Black women with varying levels of post-secondary education may often be the only or one of few employees from minoritized or marginalized groups [47,48]. Consequently, these Black women may endure higher levels of stress, lower emotional well-being, and impaired functioning due to discrimination, tokenism, and code switching to assimilate with their workplace environment [48]. Ultimately, self-report questionnaires need to consider a more nuanced approach to assess the protective potential of cultural/racial salience on the maternal functioning and well-being of postpartum Black women. 

### 4.4. Limitations and Strengths

There are limitations with the study that should be considered when evaluating the findings. Although there was a sufficient sample size to find significant findings, some of the racial identity cluster sizes were small and some of the categories for other characteristics assessed had to be combined. Secondly, the representativeness of the sample may also be limited due to sampling bias because of the restrictions put into place during the coronavirus pandemic, which affected the recruitment strategies of the original study [1]. Most women in the study were in their late twenties and had some level of post-secondary education, and lived in Georgia, factors that limit the generalization of findings. Although the MAI allowed participants to self-report the nature of attachment with their infant during a time when in-person interaction was limited, the Cronbach’s alpha of the MAI (0.93), may indicate redundancy in the items due to testing the same question in various ways [49]. Future research should incorporate observational assessment tools to understand the maternal–infant attachment between a mother and infant instead of relying on a self-report tool. Further, future research using a qualitative design may provide more insight as to how Black women’s racial identities are related to their mental health before, during, and after pregnancy. The findings could guide clinicians and mental health specialists on ways to assist Black women in navigating motherhood in a racialized society.

## 5. Conclusions

This work provides new evidence regarding the role of various clinical and racial factors on Black postpartum women’s adjustment to motherhood. In this analysis, multivariate models revealed that collectively, Black racial identity, PDS, EPDS, maternal–infant attachment, and level of education were important factors to the maternal functioning of Black postpartum people. While further research is needed to increase generalizability and better understand the relationships between these concepts, this research has potential to impact clinical practice. These findings could have direct implications for mental health providers and occupational therapists whose primary clinical priority is the improvement of mental health/functioning. These findings may have special significance regarding prevention and could potentially inform programming for organizations that support pregnant and postpartum women.

## Figures and Tables

**Table 1 jcm-12-00647-t001:** Maternal functioning and clinical characteristics of Black postpartum mothers living in Georgia, February–April 2020 (*n* = 116).

Characteristics	Mean (±SD)	Median (Range)
BIMF Score	97.4 (±13.3)	98.0 (42.0–120.0)
EPDS Score	7.1 (±5.3)	7.0 (0.0–30.0)
MAI Score	100.3 (±6.8)	103.0 (51.0–104.0)

BIMF = Barkin Index of Maternal Functioning; EPDS = Edinburgh Postnatal Depression Scale; MAI = Maternal Attachment Inventory.

**Table 2 jcm-12-00647-t002:** Bivariate analyses to assess associations between characteristics of Black postpartum mothers living in Georgia and maternal functioning.

	BIMF			
**Characteristic**	** *n* **	**Mean (±SD)**	**Median (Range)**	***p*-Value ^a^**
Racial Identity Clusters				**0.0012 ^b^**
Low Race Salience	5	103.4 (±8.4)	101.0 (95.0–117.0)	
Assimilated and Miseducated	19	105.7 (±10.6)	106.0 (77.0–120.0)	
Self-Hating	12	86.1 (±18.1)	89.5 (42.0–115.0)	
Anti-White	25	93.8 (±9.6)	94.0 (78.0–118.0)	
Multiculturalist	42	99.0 (±10.5)	99.0 (78.0–117.0)	
Conflicted	13	95.5 (±18.8)	95.0 (57.0–120.0)	
Type of Delivery for Youngest Child				0.58
Vaginal	72	98.1 (±13.1)	98.5 (57.0–120.0)	
Caesarean	44	96.4 (±13.7)	97.5 (42.0–117.0)	
Current Relationship Status				0.49
Married/Living with Partner	82	96.8 (±13.7)	98.0 (42.0–120.0)	
Single	34	99.1 (±12.4)	99.5 (75.0–120.0)	
Level of Education				**0.015 ^c^**
High School Diploma/GED or Less	21	104.3 (±10.6)	106.0 (79.0–117.0)	
Some College/Tech School/Associate’s Degree	40	94.9 (±13.9)	95.0 (42.0–120.0)	
Bachelor’s Degree or Higher	55	96.7 (±13.1)	96.0 (57.0–120.0)	
Current Employment Status				0.25
Unemployed/Stay-at-Home Mom	36	100.7 (±12.4)	103.0 (75.0–120.0)	
Part-Time	14	97.4 (±11.7)	96.5 (77.0–116.0)	
Full-Time	66	95.7 (±13.9)	95.5 (42.0–120.0)	
Type of Health Insurance				0.96
None/Self-pay	32	97.1 (±16.5)	101.5 (42.0–120.0)	
Medicaid/Medicare	64	97.6 (±12.0)	95.5 (57.0–120.0)	
Private	20	97.4 (±12.3)	99.0 (75.0–118.0)	
Household Income in the Past Year				0.18
< USD 26,000	41	99.4 (±15.0)	103.0 (42.0–120.0)	
USD 26,000-USD 74,999	45	95.2 (±13.1)	95.0 (57.0–118.0)	
≥ USD 75,000	30	98.1 (±10.7)	97.0 (78.0–120.0)	
	** *n* **	**rho**		***p*-value ^d^**
EPDS Score	116	−0.48		**<0.0001**
MAI Score	116	0.38		**<0.0001**
Mother’s Age (years)	116	−0.06		0.5
Youngest Child’s Age (months)	116	−0.01		0.91
No. of Dependent Children	116	−0.01		0.89

Bold *p* values are statistically significant. ^a^ Derived from the Mann–Whitney U or Kruskal–Wallis test. ^b^ Assimilated and Miseducated Cluster significantly different than Self-Hating Cluster and Anti-White Cluster using the Dwass, Steel, Critchlow–Fligner multiple comparison test. ^c^ HS/GED or less group significantly different than Some College/Technical School/Associate’s Degree group and Bachelor’s Degree or Higher group using the Dwass, Steel, Critchlow–Fligner multiple comparison test. ^d^ Derived from Spearman’s Rank–Order Correlation.

**Table 3 jcm-12-00647-t003:** Multivariate analyses to assess factors independently associated with the maternal functioning (*n* = 116).

	BIMF; *R*^2^ = 0.5581		
Factor	β	Standard Error	*p*
Racial Identity Clusters (ref. = Assimilated and Miseducated)			**0.0238 ^a^**
Low Race Salience	−1.17	4.73	
Self-Hating	−11.13	3.53	
Anti-White	−7.56	2.97	
Multiculturalist	−4.43	2.77	
Conflicted	−2.69	3.48	
EPDS Score	−1.13	0.17	**<0.0001**
MAI Score	0.67	0.14	**<0.0001**
Level of Education (ref. = High School Diploma/GED or Less)			**0.0342 ^b^**
Some College/Tech School/Associate’s Degree	−6.72	2.60	
Bachelor’s Degree or Higher	−3.65	2.68	

GED = General Educational Development; BIMF = Barkin Index of Maternal Functioning; EPDS = Edinburgh Postnatal Depression Scale; MAI = Maternal Attachment Inventory. ^a^ Significant differences between Assimilated and Miseducated and Anti-White Clusters and Assimilated and Miseducated and Self-Hating Clusters. ^b^ Significant difference between High School Diploma/GED or Less and Some College/Tech School/Associate’s Degree. Bold *p* values are statistically significant.

## Data Availability

The data presented in this study are available on request from the corresponding author.

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
