# Peer review of "Factors Associated with Postpartum Maternal Functioning in Black Women: A Secondary Analysis"

_jcm, 2023, doi:10.3390/jcm12020647_

Round 1
Reviewer 1 Report
The authors are exploring Black women’s experience with postpartum depression through an intersectional lens. The authors assess the associations between maternal functioning and Black racial identity, along with additional characteristics, were assessed with bivariate and multivariate analysis. The manuscript is well written, fills an important gap in the literature, and has the potential to help clinicians develop holistic strategies to better support Black women postpartum. However, there is room for improvement.
In the abstract the authors abbreviate “US” in the first sentence, then spell it out then abbreviate again. It is typical to not abbreviate in an abstract. On line 145 the authors abbreviate the United States as USA. On line157 the authors spell out the BIMF then abbreviate. This has already been done above There is a lack of consistency with abbreviations throughout the manuscript. Line 248 there is a typo.
In the discussion, the authors discuss that women who de-emphasize their Black identity may allow them to avoid distress symptoms. I would re-word this sentence. The discussion on maternal functioning and levels of education is very well written and interesting. The authors do not provide any policy suggestions for future research suggestions. I would encourage that authors to create a future research paragraph or at least include a few sentences on this topic. A qualitative analysis with some of the participants in the study would be very insightful since there are so many intersecting issues being examined.
Author Response
Thanks so much for your thoughtful review. Please see the attachment as to how we have addressed/incorporated your feedback.
Reviewer 2 Report
Thank you for the opportunity to review "Factors Associated with Postpartum Maternal Functioning in Black Women submitted to the Journal of Clinical Medicine." Let me state, the study is interesting and addresses a narrow gap in the literature with a small sample from Georgia. The manuscript will be a good addition to the literature once several limitations are addressed and some areas are revised for clarity. For this reason, there is a marked pdf attached with extensive feedback and the summary comments in the next section. Please keep in mind, the feedback in the pdf is very critical but intended to be constructive to improve the overall quality of the manuscript. Good luck with the revision process.
-----SUMMARY COMMENTS-----
As this is a secondary data analysis from a cross-sectional study with a small sample (n=116) of African American women, the title should be changed to be consistent with the reporting guidelines (e.g. STROBE).
Factors Associated with Postpartum Maternal Functioning in African American Women: A Secondary Data Analysis.
Thank you for stating this to be a secondary data analysis, but this needs to be clearly stated as recommended in the attached manuscript. Also, the sampling specifically focused on African American women rather than Black women. For this reason, the language in the manuscript needs to be consistent with this observation.
The introduction needs to be revised to clearly present the theoretical/conceptual model/framework. There are multiple concepts stated in the introduction without a clear alignment to the purpose of the study. Notably, the hypothesis for the previous study were clearly stated but I was unable to map the a prior relationships for this study. For this reason, the introduction needs to present the concepts of interest aligned with the research question (or purpose statement) and operationalized with the instruments. In this regard, a figure or diagram might be helpful for presenting this critical information for the reader. The authors appropriately noted intersectionality in the introduction without really explain the relevance to the study (this seems to be the foundation). In this regard, I would expect to see citations from this literature (e.g. Crenshaw, Collins, Greaves, Leath, Serrant-Green). Also, the discussion about instruments is distracting, and seems partially redundant with the methods section. The introduction might be better organization with the introduction (two paragraphs about the topic) followed by a background with subheadings to present the major concepts. This is merely a recommendation.
The methods section needs to be shorted as this is a secondary data analysis rather than a study with an original data collection. For this reason, there are multiple comments and recommendations provided in the attached document. The Cronbach alpha should be presented for each instrument. Also, the psychometric properties for each instrument specific to African American women should be provided, if available. This was the case for one instrument but not the others. Again, please make sure the instruments operationalize the concepts in the purpose statement. The data analysis section is weak as the process for the multivariate analysis was not clearly described. Without the clearly defined hypothesis driving the data analysis, the process could be perceived to be data mining which doesn't seem to be the case.
The results section has too much repetitive information about the sample. In a secondary data analysis, this information is usually presented in the sample area rather than the results. Also, the information that is not statistically significant or not related to the hypothesis needs to be removed from the tables. The tables have too much information that overlaps with the previous publication, and not relevant to the findings.
There needs to be some clarity about the relevance of Sojourner Syndrome to the current study. The section in the discussion is very important but needs to directly address the findings with the proper referencing. There are also major links to intersectionality in multiple literatures, including anthropology, sociology, and feminism. In this regard, none of the seminal works specific to the health disparities and health outcomes resulting from the syndrome are addressed in the section.
The conclusion needs to be revised to be clear, concise, and to the point about the findings, including practice implications and/or research. There seems to be some strong statements approaching causal relationships that might not be correct. These are noted in the attached pdf.
Finally, please provide the exempt determination number and date from the IRB for the secondary data analysis.

Author Response

(The authors gave the same response as above.)
